# Mantle hydration and the role of water in the generation of large igneous provinces

Jia Liu[1,2], Qun-Ke Xia [1], Takeshi Kuritani[3], Eero Hanski[4] & Hao-Ran Yu[2]

The genesis of large igneous provinces (LIP) is controlled by multiple factors including anomalous mantle temperatures, the presence of fusible fertile components and volatiles in the mantle source, and the extent of decompression. The lack of a comprehensive examination of all these factors in one specific LIP makes the mantle plume model debatable. Here, we report estimates of the water content in picrites from the Emeishan LIP in southwestern China. Although these picrites display an island arc-like $H_2O$ content (up to 3.4 by weight percent), the trace element characteristics do not support a subduction zone setting but point to a hydrous reservoir in the deep mantle. Combining with previous studies, we propose that hydrous and hot plumes occasionally appeared in the Phanerozoic era to produce continental LIPs (e.g., Tarim, Siberian Trap, Karoo). The wide sampling of hydrous reservoirs in the deep mantle by mantle plumes thus indicates that the Earth's interior is largely hydrated.

[1] School of Earth Sciences, Zhejiang University, Hangzhou 310027, China. [2] School of Earth and Space Sciences, University of Science and Technology of China, Hefei 230026, China. [3] Graduate School of Science, Hokkaido University, Sapporo 060-0810, Japan. [4] Oulu Mining School, University of Oulu, P. O. Box 3000 90014 Oulu, Finland. Correspondence and requests for materials should be addressed to J.L. (email: liujia08@ustc.edu.cn) or to Q.-K.X. (email: qkxia@zju.edu.cn)

Multiple physical and chemical factors contribute to the production of large igneous provinces (LIP): abnormally high mantle temperatures, fusible components in the mantle source, a large extent of decompression and the presence of volatiles (especially $H_2O$)[1, 2]. One group of researchers suggests that the melting anomalies originate by decompressional melting of hot, deep-sourced mantle plumes[3, 4], while some others propose that they are due to the presence of high proportions of easily fusible, recycled or delaminated crustal material or a large amount of $H_2O$ in the mantle sources[5–7]. The reason for this discrepancy resides in that so far, comprehensive and independent investigations on the role of all these factors in the generation of a specific LIP are rather scarce. For instance, while the plume-like mantle potential temperature ($T_p$) and the presence of pyroxenites in the source have been suggested to play a significant role in the generation of the Siberian Traps[8], the contribution of volatiles remains ambiguous, as the primary water content of magmas may be underestimated[9]. Moreover, although the $H_2O$ content of the primary basaltic magma in the Tarim LIP has been found to be up to $4.8 \pm 1.0$ wt.%[10], the role of the mantle potential temperature and pyroxenite in the source have not been investigated simultaneously.

The Permian Emeishan LIP (ELIP), one of the largest continental LIPs in the world, is located in the western margin of the Yangze Craton, Southwest China. It has been widely suggested to be related to a mantle plume[11–13], and the mantle potential temperature ($T_p$) estimated for some picrites seems to be higher than 1700 °C under dry conditions[14]. Pyroxenite in the mantle source has been suggested to be a direct source of some picrites[15, 16]. In addition, a fast kilometre-scale crustal uplift prior the eruption of ELIP basalts, which may imply fast decompression of the mantle source, has been suggested[17], although it was recently questioned by ref. [18]. However, the role of water in the initiation of the ELIP has not been carefully examined so far. The only existing estimation of the primary water content of the ELIP, ~4.0–5.6 wt.%, is from ref. [19]. The authors investigated the water content of the primary magma (equilibrium with olivine with $Fo_{91}$) based on the correlation between $H_2O/Ce$ and Ba/La, Ba/Nb

and Rb/Nb ratios, which were calibrated for global oceanic arcs. However, this estimation still needs caution to take into account for two reasons. Firstly, this primary magma water content is similar to that of the rather evolved magma when plagioclase was among the liquidus phases ($H_2O$ ~4.4 wt.%, MgO of the bulk rock <8 wt.%); and secondly, none of the primary magmas of the ELIP show typical arc-like trace element signatures, which argues against a subduction zone setting[14].

In this work, we investigated the water content of the picrite lavas from Dali, which are located stratigraphically at the bottom of the Binchuan section in the interior zone of the ELIP, representing the initial stage of the ELIP volcanism[14] (Supplementary Fig. 1). Major and trace element analyses of melt inclusions in olivine phenocrysts and major and minor element (Ca, Mn, Ni, Al) compositions of the host mineral phases were used to constrain the source lithology and the crystallisation temperature of olivine. In addition, we compiled estimates of the water content, the proportion of pyroxenite-derived melt ($X_{px}$) and $T_p$ from other continental LIPs. These data provide us so far the most comprehensive demonstration of the controlling factors for the melting anomaly in the generation of LIPs.

## Results

**Pyroxenite-bearing mantle source.** All the studied samples are massive and show a typical porphyritic texture, with olivine, clinopyroxene and spinel being the phenocrysts (Supplementary Fig. 2). Their bulk MgO contents range from 18.6 to 24.0 wt.% (calculated on anhydrous basis, see Supplementary Data 1), which are comparable to the previous analyses on bulk rocks and melt inclusions in olivine (Supplementary Data 1, 2, Supplementary Fig. 3). The olivine grains in these samples have significantly higher NiO and Fe/Mn than those expected for olivine from global mid-ocean ridge basalt (MORB) (Supplementary Data 1, Supplementary Fig. 4), suggesting a contribution of melts derived from a pyroxenitic source[20]. On the basis of the Fe/Mn ratios of high-Fo olivines and the method of ref. [20], the amount of pyroxenite-derived melts for the Dali picrites is estimated to be

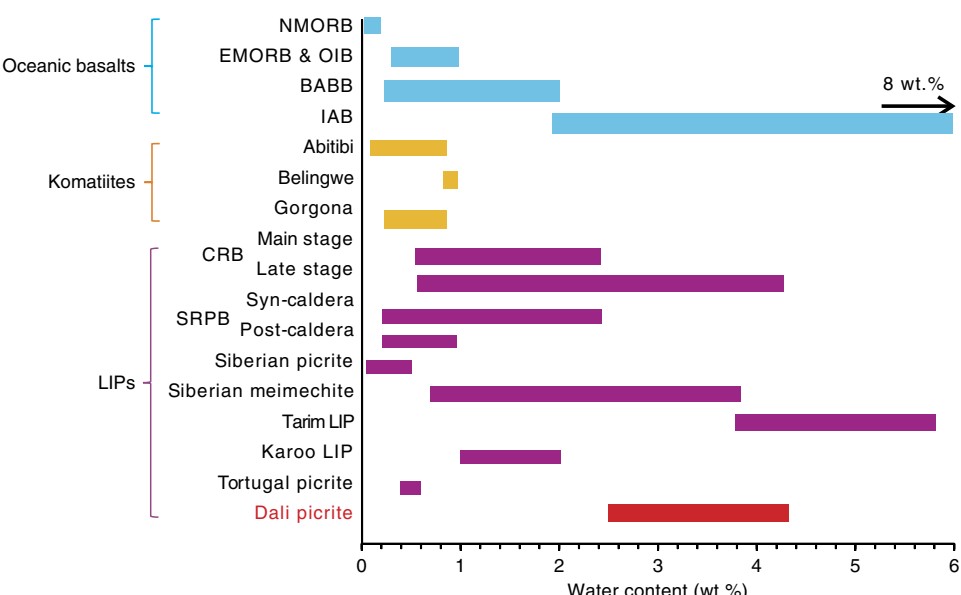

**Fig. 1** Comparison of water contents in the Dali picrites and other primary magmas from different geological settings. The water abundance data for oceanic basalts and komatiites are from refs. [26–28, 45], and the data source for other LIPs are given in Supplementary Data 3. The Dali picrite data are from this study. NMORB, normal mid-ocean ridge basalt; EMORB, enriched mid-ocean ridge basalt; OIB, oceanic island basalt; BABB, back-arc basin basalt; CRB, Columbia River basalt; SRPB, Snake River Plain basalt

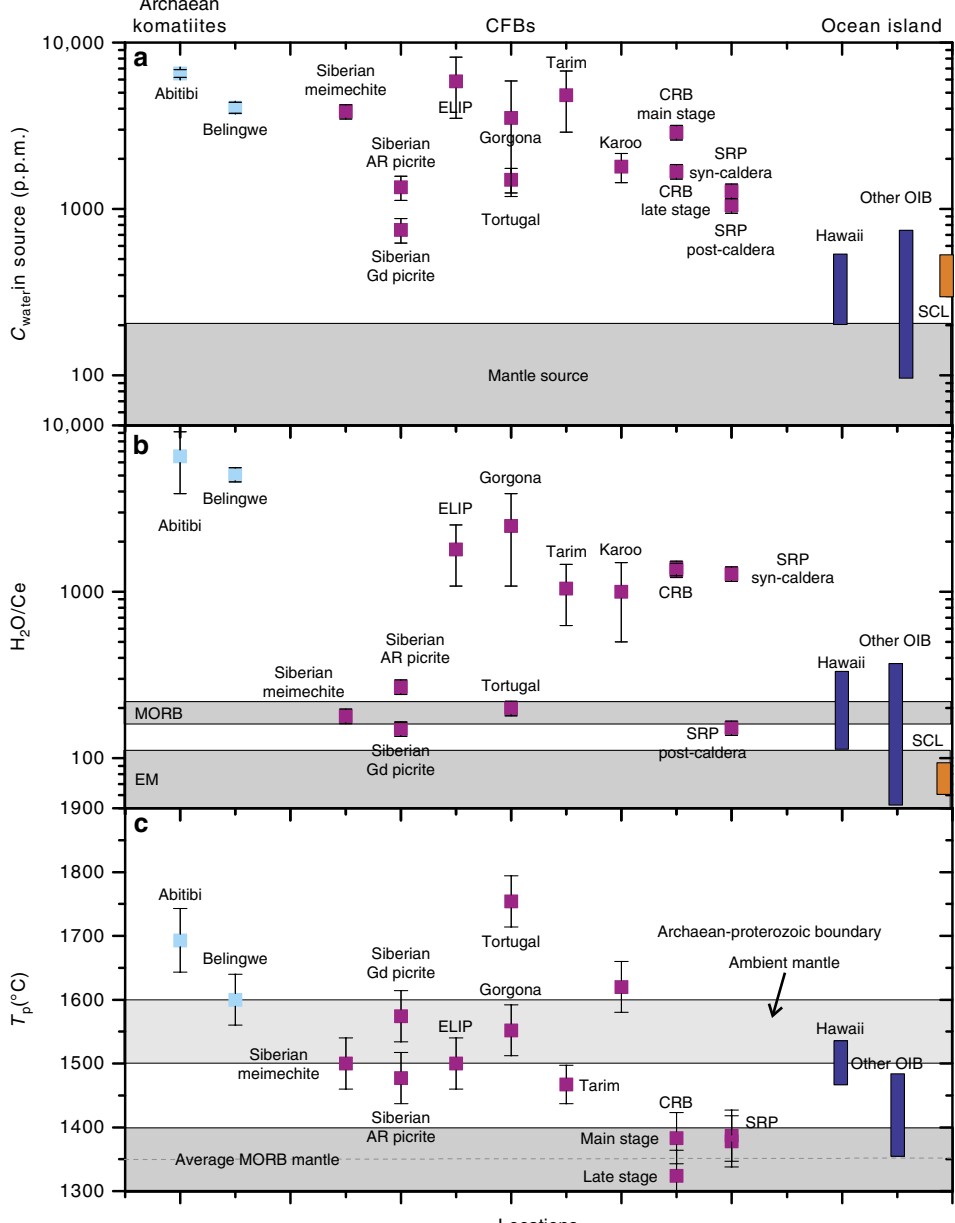

**Fig. 2** Comparison of source water contents, $H_2O/Ce$ ratios and estimated maximum mantle potential temperatures for the sources of komatiites, LIPs and OIBs. **a** The water content in the sources were calculated from the $H_2O$ content of the primary magma (the data source for the komatiite and LIPs are the same as in Fig. 1), assumed degree of partial melting, and the partitioning coefficient of water between mantle rock (peridotite or pyroxenite) and magma. **b** See the Supplementary file for the details of calculation of $H_2O/Ce$ ratios. **c** All the $T_p$ have been corrected for the effect of water, see Methods for the calculation and Supplementary Data 3 for results. The $T_p$ range for the Archaean-Proterozoic boundary and the average MORB mantle are from refs. [6, 34,]. All the error bars represent 2 standard deviation. Gorgona komatiite and Tortugal picrite are from Caribbean LIP (CLIP); Siberian AR picrite, the Ayan River picrite, Gd picrite, the Gudchikhinsky picrite; SLC, the pyroxenite xenoliths hosted in the Salt Lake Crater vent, Hawaii, data from ref. [38]. The water content and $H_2O/Ce$ ratios of the source for Hawaii and other OIBs are from ref. [38] and the literature therein. The $T_p$ of Hawaii and other OIBs are from ref. [4], after correction for the effect of water based on ref. [40]. CRB Columbia River basalts, SRPB Snake River Plain basalts

~40–50 wt.% (Supplementary Data 1). This is consistent with the recent results from melt inclusion studies: the CaO content in the primary magmas of the Dali picrites increases slightly with decreasing $TiO_2$, which requires a significant role of pyroxenite in the source[16].

**Island Arc-like water content of the primary magma**. The $H_2O$ content of the magma was recovered from the $H_2O$ content of clinopyroxene (cpx) phenocrysts and the partition coefficient of $H_2O$ between cpx and melt ($D_{water}^{cpx/melt}$)[21–23]. The $H_2O$ content of

the primary Dali picrite was estimated to be $3.44 \pm 0.89$ wt.% (See Methods for details), reaching the range of island arc basalts (IAB) (Fig. 1). Even if the uncertainty (<40%) is taken into account, the lower-limit $H_2O$ content is still as high as 2.04 wt.%.

## Discussion

The bulk-rock and melt inclusion compositions show considerable positive Pb anomalies (Supplementary Fig. 3b), which may point to significant contamination with continental crust. However, the following observations argue against this possibility:

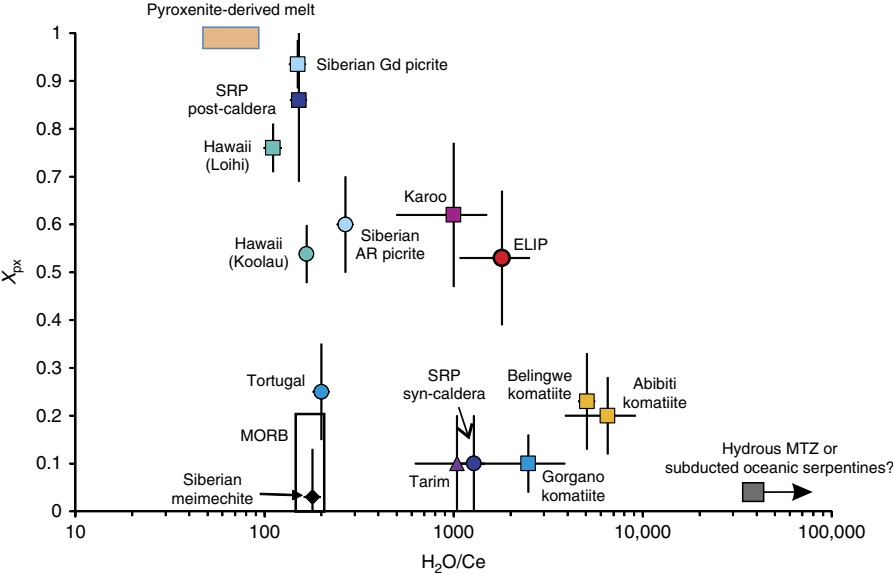

**Fig. 3** Correlation between $H_2O/Ce$ and proportion of pyroxenite-derived melts in primary magmas of LIPs and komatiites. The $X_{px}$ values were calculated based on the Fe/Mn ratios of olivine phenocrysts (Supplementary Data 1, 3). The error bars for $X_{px}$ and $H_2O/Ce$ represent 1 and 2 standard deviations, respectively. $H_2O/Ce$ ratios of global MORB and pyroxenite xenoliths from Hawaii (SCL)[38] are also shown for comparison.The $H_2O/Ce$ ratio of pyroxenite-derived melt is equal to that of the source pyroxenite if the degree of partial melting is larger than 10%, which is commonly reached in mantle plume regimes[20, 30]. $H_2O/Ce$ of the potential melts from less fertile but hydrous reservoirs in the deep mantle, mantle transition zone (MTZ) peridotite or subducted serpentinised peridotites[26, 39] are shown as the green box. $X_{px}$ of Hawaiian picrite and MORB were calculated using the data from[20]. See Supplementary Data 3 for the calculation of $X_{px}$ and $H_2O/Ce$. Note that the error bars of $H_2O/Ce$ for some samples are smaller than the size of the symbol. Gd, Gudchikhinsky; SRP, Snake River Plain

firstly, the bulk-rock MgO content and trace element characteristics of our samples are comparable to those of the melt inclusions hosted by the most magnesian olivines (Supplementary Data 2, Supplementary Fig. 3b); secondly, the plots of MgO vs. Nb/La and Ti/Y vs. Ce/Pb do not favour continental crust as a contaminant (Supplementary Fig. 5); thirdly, the Dali picrites have initial $\varepsilon_{Nd}$ values close to zero[13]. Thus, the island arc-like $H_2O$ contents of the Dali picrites should be a characteristic feature of the primary magma. Although the interaction between primary picritic melts and melts derived from the sub-continental lithospheric mantle (SCLM) has been suggested[11], the high water content in our samples cannot be explained by this mechanism, as the $H_2O/Ce$ ratios of SCLM-derived melts are much lower than that of our samples (<700 for lamprophyres and <250 for the partial melts of amphibole-rich veins[24, 25] vs. ~1800 in our samples as calculated from the estimated water content and the bulk-rock Ce concentrations). Therefore, we consider that the high $H_2O$ content in the Dali picrite magmas was derived from the sub-lithospheric mantle. Such hydrous mantle reservoirs have also been proposed for Archaean and Cretaceous komatiites[26–28]. Assuming the partition coefficient of $H_2O$ between mantle rocks (peridotite, pyroxenite, eclogite) and melt to be ~0.01[29] and the lower limit of the degree of partial melting around 15–25%[30], to form a primary picritic magma with 3.44 wt.% of $H_2O$, the $H_2O$ content of the mantle source should be >6000 p.p.m., regardless of the melting model (batch or fractional melting).

Crystallisation temperatures of olivine phenocrysts were obtained by two geothermometers based on the Fe/Mg partitioning between olivine and melt and the Al partitioning between olivine and coexisting spinel (Methods). By the Fe/Mg thermometer, the calculated crystallisation temperatures of the olivine with maximum Fo in each sample gradually decreases with decreasing Fo, and the highest obtained olivine liquidus temperature is ~1575 ± 40 °C (Supplementary Fig. 6), if the magma is assumed to be dry. When the effects of $H_2O$ content are

considered, the corrected highest temperature is ~1460 ± 43 °C (Supplementary Fig. 6). The Al thermometer gives a similar estimate for the maximum olivine liquidus temperature (Supplementary Fig. 6). It is not easy to accurately assess the mantle potential temperature ($T_p$) due to the complexity caused by the incorporation of volatiles and pyroxenite in the mantle source. Recently, Trela et al.[31] applied the relationship between the olivine liquidus temperature and $T_p$ for the peridotite system (equation 17 in ref. [32]) to estimate the mantle potential temperature for many LIPs. Here, we test this approach by applying it to a series of pyroxenite partial melting experiments (Methods). The results show that it can recover $T_p$ within an uncertainty of ±70 °C for most of these experimental runs (Supplementary Fig. 10). Thus, according to the estimated olivine solidus temperature (~1460 ± 43 °C), we estimate the upper limit of the $T_p$ to be around ~1500 ± 70 °C (Methods), which is considerably higher than that of the average MORB source. The large (40–50%) proportion of pyroxenite-derived melt coupled with an island arc-like $H_2O$ content and elevated $T_p$ indicates that the ELIP was formed by an extensively hydrated mantle plume rich in fusible components.

As shown in Figs. 1 and 2, for the Siberian Traps, Karoo LIP, Tarim LIP, Yellowstone hotspot track (Columbian River basalt and Snake River Plain basalt) and Caribbean LIP, the $H_2O$ content of their primary magma vary from 0.25 to 4.8 wt.%. Thus, the mantle source of many other Phanerozoic LIPs also seems to have been extensively hydrated. The calculated $H_2O$ content and $H_2O/Ce$ ratios of the mantle sources are from 900 to >6000 p.p.m. and ~160–400 to >2000 (Supplementary Data 3), respectively, ie, significantly higher than those of the MORB and oceanic island basalt (OIB) sources (Fig. 2a, b) and falling in the range of the IAB sources. However, their trace element signatures do not show obvious depletions in Nb and Ta (Supplementary Fig. 7), arguing against a direct contribution of the fluids released from a subduction zone. The $T_p$ after correction for the depression effect on

the liquidus temperature by $H_2O$ varies from that of an average MORB to higher than that of most of the OIBs (see Method, Fig. 2c, Supplementary Data 3).

In Fig. 3, the $H_2O/Ce$ ratios and the possible proportion of pyroxenite-derived melts in their primary magma ($X_{px}$, which is directly related to the Mn/Fe ratio of olivine, see Methods for details) were compared. For many continental LIPs that were generated by hydrous and pyroxenite-involved mantle plumes, the $X_{px}$ values correlate negatively with the $H_2O/Ce$ ratios (Fig. 3). It should be noted that although the variation in Fe/Mn ratios of olivine can be alternatively explained by partial melting of a peridotite source under different $P-T$ conditions (at higher $P-T$, the partial melts would have a higher MgO content, and thus the olivine-melt partition coefficient of Mn becomes lower leading to lower MnO and higher Fe/Mn in crystallised olivine, refs. [33, 34,]), the heterogeneity in $X_{px}$ shown in Fig. 3 reflects the fertile nature (proportion of pyroxenite) of their mantle sources. This is supported by the following observations: firstly, the Tortugal lavas in the Galapágos plume have been suggested to be results of partial melting of a peridotite source at ~7 GPa and ~1800 °C[31], which should have much higher Ni contents and higher Fe/Mn ratios than MORBs according to the suggestion of ref. [33]; however, as shown in Fig. 3, the $X_{px}$ values of the Tortugal olivine are comparable to those of MORB; secondly, the lithospheric mantle for the Loihi and Koolau lavas in Hawaii and the post-caldera and syn-caldera lavas in the Snake River Plain has a similar thickness, but the lavas show distinct calculated $X_{px}$ (Fig. 3). Actually, the recent investigation on the trace element concentration in olivines from the Tuli and Mwenezi picrites in the Karoo Continental Flood Basalt (CFB) and the Horingbaai low-Ti picrites in the Etendeka CFB, all of which erupted in regions of thickened (>90 km) lithosphere in southern Africa, shows that Karoo picrite olivines have systematically higher Fe/Mn ratios and higher Ni contents than Etendeka picrites[35]. The other factor affecting the Fe/Mn ratios of the crystallised olivine would be the oxygen fugacity ($fO_2$) of the magma[36]. However, the difference in $fO_2$ could not explain the observed variation in $X_{px}$. The Siberian meichechite has a considerably higher $H_2O$ content and $fO_2$ than the Gudchikhinsky formation picrite (3.88 wt.% vs. 0.25 wt.% $H_2O$, +1.5 vs. −1.5 to −2 ΔQFM)[8, 37], while its $X_{px}$ is much lower than that of the latter (0.03 vs. 0.94). Thus, we suggest that the calculated $X_{px}$ values in Fig. 3 reflect the source heterogeneity or at least they can be considered as the maximum estimate of the pyroxenite-derived melt.

The above discussion indicates that, except for the depleted-MORB mantle (DMM), two end-members are responsible for $H_2O/Ce$ in the primary magmas in LIPs. One is pyroxenite, which is more fusible than peridotite and contains a higher $H_2O$ content but lower $H_2O/Ce$ than the DMM[38]. The other is a reservoir in the deep mantle, which is extensively hydrated and has a major element composition close to that of peridotite. This kind of mantle reservoir has been deduced for several hydrous Archaean and Cretaceous komatiites[26–28] (Fig. 3). This end-member could be the solid segments or partial melts of wadzlyeite or ringwoodite in the MTZ or the recycled serpentinised peridotites[26, 28, 39]. Although the proportion of pyroxenite-derived melts could be affected by the thickness of the lithosphere and the degree of melting of pyroxenite, the complementary contribution of this hydrous reservoir and pyroxenite in the genesis of a LIP could also be explained by that when the water in the hydrous plume for a LIP mainly resides in the less fertile and fusible part (low $X_{px}$)[26, 28], it starts melting earlier and preferentially than pyroxenite. If so, the decreasing proportion of pyroxenite-derived melts during a short time period does not necessarily mean a catastrophic thinning of the lithosphere[8], but would be due to the variation of the $H_2O$ content in the mantle plume (Fig. 3).

Overall, we propose that the mantle sources of the Phanerozoic LIPs are generally extensively hydrated and contain abundant fusible components, which, together with a higher mantle potential temperature[4], resulted in a much higher melt productivity and eruption rates in comparison with those of OIBs.

More broadly, our studies lead to new perspectives for the ancient water recycling into the deep Earth. The non-arc-like hydrous mantle represented by the Archaean komatiites (ie, Abitibi and Belingwe) has been explained by the incorporation of a hydrous reservoir from the deep mantle, the hydrous MTZ[26] or 'carbonated wetspots' formed by the early ingassing of surface water into the deep Earth in the Hadean[40]. The widespread komatiite production at 2.7 Ga has been regarded as the consequence of outgassing of this substantial volatile-rich mantle reservoir and the decline of komatiite eruptions after this time as a reflection of purging out of volatiles from the mantle[40]. The general hydration of mantle plume sources of several Phanerozoic LIPs that were not related to the direct involvement of fluids released in a subduction zone (Supplementary Fig. 6) indicate that the Earth's interior during the Phanerozoic was still occasionally extensively hydrated. One possible water reservoir would be the MTZ, which have been demonstrated to be regionally hydrous[41]. Alternatively, the water was entrained by the mantle plume derived from the large low-shear-velocity provinces (LLSVPs), which is a unique thermochemical structure in the lower mantle and would probably also be modified/incorporated by the volatile elements transported into the deep Earth by residual subducted plates[42]. The regassing of the deep mantle would be explained by that, at least before the end of the Proterozoic, the mantle had cooled enough for the water to be carried into the MTZ or the lower mantle through the subduction of oceanic lithosphere[43]. This is comparable to the recent suggestion by Korenaga et al.[44] that there is a net water transport from the oceans to the mantle, with a long-term (at least back to ~3 Ga) average rate of $3–4.5 \times 10^{18}$ g per year, which is based on the most complete modelling of the continental freeboard so far.

## Methods

**Major and trace element analysis of bulk-rock samples**. After the weathered surfaces were removed, the samples were crushed into small fragments (<0.5 cm in diameter). Then, they were cleaned with deionised water and ground to 200-mesh powder. Major and trace element concentrations of bulk-rock samples were analysed at ALS Chemex (Guangzhou, China) Co., Ltd. Loss-on-ignition was determined before X-ray fluorescence (XRF) analysis. Major element compositions were measured by XRF spectrometry on fused glass disks. The precision for element concentrations of >1 wt.% was 1–3% and ~10% for element concentrations of <1 wt.%. For trace elements, two methods were employed to check the accuracy of the analytical results against each other. In the first method, lithium metaborate flux was added to the sample powder, and the mixture was fused at 1000 °C and dissolved in 100 ml of 4% nitric acid. The powder was then analysed using a PerkinElmer inductively coupled plasma mass spectrometer (ICP-MS). In the second method, the sample powder was dissolved in a Teflon bomb using an $HNO_3$-HF-$HClO_4$ mixture, diluted by dilute nitric acid after being dried on a hot plate and then analysed by ICP-MS. The precision is generally better than 5% for most trace elements. These two methods gave rather consistent results with 95% confidence.

**Electron microprobe analyses**. Chemical compositions of olivine and clinopyroxene phenocrysts and chrome spinel hosted by olivine were measured by a Shimadzu electron probe microanalyzer (EPMA 1600) at the CAS Key Laboratory of Crust-Mantle Materials and Environments (CA-CMME) of the University of Science and Technology of China (USTC). Back-scattered electron images were used to check the homogeneity of the minerals. During the quantitative analysis of clinopyroxene (cpx) phenocrysts and spinel inclusions in olivine, the operating conditions were the following: 15 kV accelerating voltage, 20 nA beam current and 1 μm spot size. Only the spinel grains located in the central zone of the host olivine phenocrysts were analysed. For olivine, the analytical method described by ref. [19] was used, employing an accelerating voltage of 20 kV, a beam current of 300 nA and a spot size of 5 μm. Natural minerals and synthetic oxides were used as standards, and a programme based on the ZAF procedure was used for all data correction. All the analysed points in cpx phenocrysts were set within thefourier-

transform infrared spectroscopy (FTIR) analysis region. Some electron probe micro-analyses (EPMA) of cpx were conducted with a JEOL electron probe microanalyzer in the Hefei University of Technology. Analytical conditions were similar to those used in USTC.

**Major and trace element analysis of melt inclusions.** Major elements of the melt inclusions and their host olivine grains were analysed using a JEOL Superprobe JXA-8200 electron microprobe at the Max Planck Institute for Chemistry, Mainz, Germany. The analytical conditions for were the following: 15 kV accelerating voltage, 12 nA electron beam current and defocused to 5-µm-sized beam. For olivine, the conditions were similar to those for cpx phenocrysts described above. Trace element concentrations in melt inclusions were analysed by LA(laser ablation)-ICP-MS at the University of Tasmania. This instrumentation comprises a New Wave Research UP213 Nd-YAG (213 nm) laser coupled to an Agilent 4500 quadrupole mass spectrometer. For this study, analyses were performed in a He atmosphere by ablating 20–70 µm-diameter spots at a rate of five shots per s using laser power of ~12 J/cm$^2$. See Supplementary Data 2 for the data.

**Analysis of the H$_2$O content of single clinopyroxene.** The water contents of the picritic melts from which clinopyroxene phenocrysts crystallised were calculated by the following steps:

First, the cpx/melt partition coefficient ($D_{water}^{cpx/melt}$) of water was calculated based on the major element composition data determined by EPMA (Electron probe micro-analyzer) analysis of cpx and the following equation from ref. [45]:

$$\ln D_{water}^{cpx/melt} = -4.2(\pm 0.2) + 6.5(\pm 0.5)X_{IV_{Al}}^{cpx} - 1.0(\pm 0.2)X_{Ca}^{cpx}. \qquad (1)$$

Second, the water content of the basaltic melt was calculated based on the water content in the cpx determined by FTIR and the calculated $D_{water}^{cpx/melt}$.

The water content of a single cpx phenocryst was measured following the approach of refs. [21, 46], in which the unpolarised fourier-transform infrared spectroscopy (FTIR) beam was focussed on a randomly orientated thin slices of cpx. Both theoretical calculations and experimental tests[21, 46] have shown that this method allows an estimation of the water content in those cpx grains that have three groups of OH bands (~3640 cm$^{-1}$, 3640 cm$^{-1}$, ~3460 cm$^{-1}$) and the highest linear absorption of OH bands (the height of the peak) <0.3, within 20% difference compared with the classical approach conducted with polarised IR (need to cut the crystals along specific principal directions; ref. [47]). Combining the uncertainty of the OH absorption coefficients and the base line correction, the total uncertainty in the water content measured by this method is <30%[21], which is comparable to the uncertainty in the secondary iron mass spectrometer measurements[20]. Considering the typical uncertainty of the water partitioning coefficient predicted by the equation 10 of O'Leary et al.[46] (~20%), the total uncertainty of the calculated water content in the melt is around 40%[46]. All the cpx phenocrysts in this study contain these three groups of OH bands (Supplementary Fig. 8a), and their highest linear absorptions are less than 0.3, allowing us to apply such an unpolarised method confidently. See the Supplementary Information for the FTIR spectrum of OH in cpx phenocrysts, the evaluation of possible H diffusion in cpx and the choice of cpx to calculate the final water content of magma.

**FTIR analysis.** The unpolarised FTIR measurements were conducted at the CAS Key Laboratory of Crust-Mantle Materials and Environments of the University of Science and Technology of China (USTC) with a Nicolet 5700 FTIR spectrometer coupled with a Continuµm microscope using a KBr beam splitter and equipped with a liquid-nitrogen-cooled MCT-A detector. Double-polished thin sections of picrite samples were prepared with thicknesses ranging from 120 to 150 µm. The light source and pathway, and the sample-holding cell were flushed with a purified air free of CO$_2$ and H$_2$O. For each cpx grain, unpolarised spectra ranging from 1000 to 4500 cm$^{-1}$ were obtained using a total of 256 scans at a resolution of 4 cm$^{-1}$. A square light spot with dimensions ranging from 30 by 30 µm to 100 by 100 µm was used, depending on the size and quality of the mineral grain. The optically clean, inclusion- and crack-free areas were chosen for measurement. A typical unpolarised IR spectrum of the OH bands in cpx is shown in Supplementary Fig. 8a. Profile analyses for a subset of cpx grains were also conducted to check the diffusion of H (Supplementary Fig. 9). The water content of cpx phenocrysts was calculated by the transformed Beer–Lambert law: $C = 3 A / (I \times t)$, where $C$ is the water content of cpx in ppm, $A$ is the unpolarised integral absorbance, $I$ is the absorption coefficient (7.09 ppm$^{-1}$ cm$^{-2}$, ref. [48]) and $t$ is the thickness in cm.

**Calculation of the water content of primary magma.** Typical FTIR bands of OH in cpx phenocrysts from the Dali picrites are shown in Supplementary Fig. 8a. These bands are rather consistent with that in mantle-derived cpx and cpx megacrysts in kimberlite[48, 49]. Results of profile analyses are displayed in Supplementary Fig. 9.

The $C_{water}^{melt}$ (water in melt) vs. Mg# of cpx plot shown in Supplementary Fig. 8d indicates a variable melt water content from ~1 to >4 wt.%. at all Mg# values. The possible reasons for this large variation include: isobaric crystallisation of

phenocrysts, which causes an increase in $C_{water}^{melt}$; degassing at depth, which reduces $C_{water}^{melt}$; and water loss through diffusion after crystallisation of cpx and emplacement of the magma in shallow crust. On the basis of the FTIR profile analyses shown in Supplementary Fig. 9, cpx in sample DL3-5 experienced obvious and prominent water loss (Supplementary Fig. 9c–f), whereas very few cpx grains in sample DL3-6 were affected by diffusion of H$_2$O (Supplementary Fig. 9a, b). Thus, the diffusional loss of water would be the reason why nearly all DL3-5 cpx grains and part of DL3-4 cpx grains contain systematically less water than the others. With the exception of these data, most of the other grains are within the range of modern island arc basalts (Supplementary Fig. 8d, light green and blue fields). Two lines of evidence indicate that the diffusion did not significantly change the water content in cpx in samples DL3-1, −4 and −6: (1) most of the profile analyses on cpx show that the water concentrations are homogenous within individual grains (Supplementary Fig. 9a, b); (2) the calculated water contents in cpx are correlated with the atom numbers of Mg and Al, which diffuse several orders of magnitudes slower than H does (Supplementary Fig. 8b, c).

In order to reduce the effects of magma crystallisation and degassing on the water content in melts, we used the data from the cpx grains with Mg# higher than 89 to calculate the water content of the 'primary' magma (Supplementary Fig. 8d), resulting in an average water content of 3.73 ± 0.89 wt.%. The maximum Mg# of cpx is slightly lower than the Fo number of olivine in the Dali picrites (91.1 and 93.4, respectively; Supplementary Data 1). Assuming that $K_D(Fe - Mg)^{cpx-liq}$ and $K_D(Fe - Mg)^{ol-liq}$ are 0.27 ± 0.03 and 0.30 ± 0.03, respectively, we can estimate that the amount of olivine crystallised before cpx is about 10%, based on the whole-rock composition of sample DL3-1. After the correction of the effect of the previous crystallisation of olivine, the water content of the primary Dali picritic magma is 3.44 ± 0.89 wt.%.

**Geothermometers for the olivine liquidus temperature.** For the Fe/Mg thermometer, the crystallisation pressure of cpx phenocrysts is calculated first according to the approach in ref. [50]. The result is around 1.3 GPa. To estimate crystallisation temperatures of olivine, we used partitioning of Al between olivine and spinel[51] and Fe/Mg distribution between olivine and melt[52]. The olivines with Fo in the range of 81.6–91.9 mol.%, crystallised together with chrome spinel, were used in the olivine-spinel geothermometry. Only the spinel grains hosted in the central part of olivine phenocrysts were chosen. Unfortunately, the olivine grain with highest Fo (93.4 mol.%) has only spinel in its rim and was not applied in this geothermometry. The error of this thermometer is estimated to be ±25 °C[51] and includes the error arising from the precision of the olivine Al$_2$O$_3$ measurement (0.003 to 0.0075 wt.%). Before the application of the Fe/Mg olivine geothermometry, the equilibrium between the olivine and the bulk-rock composition was examined. Olivine shows a wide range of Fo, and assuming $K_d$(Fe-Mg) to be 0.30 ± 0.03[53], most of them are not in equilibrium with their bulk-rock hosts. However, the maximum Fo contents of olivine are close to the equilibrium value. The PRIMELT3 software of ref. [31] was used to subtract the fraction of extra previously crystallised olivine from the whole-rock composition to obtain the liquid composition equilibrated with the observed most Fo-rich olivine. The $K_d$ values calculated based on the obtained liquid compositions and maximum Fo fall in the range of 0.30–0.33. Then the olivine- and glass-based thermometer of ref. [52] was used to calculate the olivine liquidus temperature (equation 22) with the experimentally determined water content. The standard error of this approach is around ±43 °C[52].

**Estimation of mantle potential temperature.** It is shown in this study that the mantle source of the Dali picrites was hydrous and contained pyroxenite (Fig. 1, Supplementary Fig. 5). In this case, the models of refs. [31, 54], which were calibrated from experiments on peridotites, could not be directly applied to obtain mantle potential temperatures ($T_p$). For instance, an incorrect application of the PRIMELT2 model to lavas that formed from pyroxenite sources is expected to result in an $T_p$ overestimation of ~50–70 °C[55]. Considering that we have determined the olivine liquidus temperature at a lower crust depth ($T_1^{Ol/L}$) by an independent approach (Al-in-olivine thermometer), to recover the $T_p$, we use the correlation between olivine liquidus temperature at 1 atm ($T_1^{Ol/L}$) and $T_p$ calibrated by ref. [31] for peridotite-derived magmas:

$$T_1^{Ol/L} = T_P^{Ol/L} - 54P + 2P^2, \qquad (2)$$

$$T_p = 1.049 T_1^{(Ol/L)} - 0.00019(T_1^{(Ol/L)})^2 + 1.487 \times 10^{-7}(T_1^{(Ol/L)})^3, \qquad (3)$$

where $P$ is the olivine crystallisation pressure in GPa and all the temperatures are in °C.

The errors of applying these equations to pyroxenite sources were tested for a series of partial melting experiments conducted for pyroxenite with a wide range of bulk compositions (Supplementary Data 4) at pressure of 1–7 GPa and temperatures of 1250–1750 °C. In the test, the $T_1^{Ol/L}$ was calculated by the following equation[32]:

$$T_1^{(Ol/L)} = 1020 + 24.4 MgO - 0.161 MgO^2 \qquad (4)$$

where MgO is the content for the produced partial melts. As shown in Supplementary Fig. 10a (for data, see Supplementary Data 4), the calculated mantle

potential temperature ($T_{p\text{-}Cal}$) based on the above equations can recover $T_p$ for pyroxenite partial melt experiments ($T_{p\text{-}Exp}$), most of which are within the uncertainty of $\pm 70\,°C$. The mantle potential temperature for Dali picrite is calculated as follows: (1) by Eq. (1), $T_1^{Ol/L}$ is calculated from the measured $T_1^{Ol/L}$ (~1460 °C) and crystallisation pressure of 1.3 Gpa; (2) then Eq. (2) is used to calculate $T_p$. The calculated $T_p$ for Dali picrite is ~1500 ± 70 °C. For the other LIPs, whose $T_1^{Ol/L}$ was determined by the Al-in olivine thermometer[51], Eq. (2) above is used to calculate their mantle potential temperatures. This calculation gives consistent $T_p$ estimates based on the Fe-Mg equilibrium thermometer when the effect of water is considered[25, 30]. For those without Al-in-olivine-based $T_1^{Ol/L}$, the $T_1^{Ol/L}$ values were calculated based on Eq. (3) and their primary magma MgO content, and then the water depression effect was calibrated based on the approach of[25, 56, 57], and finally $T_p$ was calculated by Eq. (2) with such a water-calibrated $T_1^{Ol/L}$. The $T_p$ values for OIBs were estimated according to the previously determined value by ref. [4] and the water depression effect by ref. [39]. See Supplementary Data 3 for the calculation results.

**Data availability**. All the data used in this study are reported in four Supplementary Data online.

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

## Acknowledgements

We thank J.L. Xu, J. Wang and Y.H. Shi for their help in the EPMA analysis. This study was supported by the Strategic Priority Research Program (B) of Chinese Academy of Sciences (grant no. XDB18000000), the National Natural Science Foundation of China (grant nos. 41630205 and 41772049) and Academy of Finland (grant no. 281859).

## Author contributions

Q.-K.X. designed and led the project. J.L. and H.-R.Y. finished all analyses except the major and trace element data of the melt inclusions that were provided by E.H. J.L. and Q.-K.X. wrote the manuscript with inputs from T.K. and E.H. All authors contributed the interpretation of the results.

## Additional information

**Competing interests:** The authors declare no competing financial interests.

