## [Peer Review File · Nature Communications]

Reviewers' comments:

Reviewer #1 (Remarks to the Author):

Traditionally, the formation of large igneous provinces (LIPs) is interpreted as the product of mantle plumes. Because of its higher mantle potential temperature than the normal mantle, upwelling of a mantle plume will cross the dry peridotite solidus at more deeper depth than that beneath the MORB mantle, thus can generate higher volumes of magmas (assuming a similar final melting depth, this will lead to a significantly wider melting interval). Petrology, isotope and trace element geochemistry of MORBs and OIBs and continental basalts show that the Earth mantle is highly heterogeneous as the consequence of recycling of crustal material by subduction or delamination into the convection mantle. For this reason, the presence of fusible fertile components (e.g., pyroxenite) and volatiles (e.g., CO₂ or H₂O) in the mantle source should also remarkably reduce the mantle solidus, therefore capable for the genesis of the LIPs. In this manuscript, Liu et al. present, to my knowledge, the first Fourier transform infrared (FTIR) spectrometry analysis of clinopyroxene phenocrysts and other geochemical results of picrites in the Emeishan LIP. The authors show that the estimated primary magmas of these picrites have very high water contents, which is similar to the island arc basalts, but significantly higher than those for MORB and OIB. Based on these results, they proposed that mixing a low H₂O/Ce pyroxenite endmember with a hydrous peridotite endmember could explain their geochemical features. At last the authors suggested the deep mantle during the Phanerozoic is widely hydrated. The H₂O data for the ELIP picrites is new, but the explanation of these data has some flaw as listed below, and thus a major revision is required before it can be accepted for publication.

λ Lines 17-18. It is puzzling that why such a high H₂O content do not affect trace element compositions of the magmas, since both island arc basalts and most CFBs with high H₂O contents usually display negative Nb-Ta anomaly (e.g., reviews by Plank et al. 2014, EPSL for IAB and by Wang et al. 2016, Lithos for CFB).

λ Lines 44-45. "A fast kilometer-scale uplift prior to eruption of ELIP basalts" is only an explanation, not an observational result. Some researchers have argued against this interpretation (e.g., Ukstins Peate. 2008, Nature Geosciences).

λ Lines 46-47. Wang et al. 2016, Lithos has independently estimated the water contents (4.0-5.6%) in the primary magmas of the ELIP picrites by using the liquid line of descent (LLD) for a large dataset of the ELIP basalts. Their results are similar to the data of the current manuscript. Therefore, these previously work should be evaluated, although the previous authors did not report direct analysis of H₂O content in the ELIP picrites.

λ Line 58. I checked the data, and I found the LOI values for these basalts are not low, ranging from 3.9% to 5.3%, thus the alteration effect should be examined.

λ Lines 60-63. Based only on the high Ni contents and high Fe/Mn ratios of olivines, the authors proposed a significant contribution of melts derived from a pyroxenite source. However, this is not always the case. Because all the melting temperature, pressure, and melt or olivine composition factors will control on the Ni contents and Fe/Mn ratios, e.g., a recently 2017 paper published in Nature Geosciences by Matzen and coauthors suggested that the high Ni contents and high Fe/Mn ratios of olivines can also be explained by peridotite melts at different temperatures and pressures. So I suggest the authors to recheck all these factors separately.

λ Lines 65-67. The authors have analyzed the chemical compositions of melt inclusion, I suggest to analyze the water content in melt inclusions of olivines as well, this can provide more convincing self-consistent results about H₂O contents in the primary magmas.

λ Lines 67-70. Because such a high H₂O content of the primary magmas should significantly decrease the partition coefficients of Ca between olivine and melt, thus the crystallized olivines should have low Ca contents (Gavrilenko, Herzberg et al. 2016, JP). However, I find the olivine Ca contents are not lowered remarkably. The author need to explain how to reconcile this discrepancy.

λ Line 71-76. The contaminated continental crust by high MgO magmas should not be the average composition of the continental crust, but the regional crust composition in the ELIP as suggested

by Sobolev et al. 2016, Nature for the Archaean Abitibi high-Mg Komatiites.

λ Lines 99-106. For estimating the mantle potential temperature, the author noted that water contents and the presence of pyroxenites in the peridotite source would significantly overestimate the calculated T_p . However, in the detailed Methods Section, the T_p were calculated using the method proposed by Herzberg and Asimow. 2015. This method is still based on dry peridotite hypothesis. The authors show that the estimated T_p and the uncertainty is ~ 1500 °C and ± 70 °C, respectively. Considering the H₂O contents in the mantle source, this value should be lower furthermore. Even so, I remained question the method for calculating T_p used by the authors if the peridotite source contains high water contents and pyroxenites.

λ Lines 139-146. Although Sobolev et al. 2016, Nature suggested a mantle plume origin for the Archaean komatiites, Herzberg. 2017, JP suggested an alternative model for Archaean komatiites. He provided detailed petrological and mass balance modeling evidence for an early Earth carbon and water cycle. Then he suggested these rocks were generated in the asthenospheric convection mantle in the presence of volatiles such as H₂O or CO₂, without the requirement for thermal mantle plume or anomalous high mantle potential temperature.

λ Line 153. I checked the reference cited here, and I found it is not related volatile elements transported into the deep mantle.

λ Line 154. Although the mantle has cooled enough, based on Al-in olivine thermometer, Trela et al. 2017, Nature Geosciences has suggested an ancient mantle reservoir sufficiently hot for the Phanerozoic Tortugal lavas.

λ Figure 3. For those mantle source with higher H₂O/Ce, their source should also have higher f_{O_2} , which equals to higher Fe³⁺ and lower Fe²⁺. This could lead to crystallization of olivines with low Fe/Mn ratios. Thus, before using olivine Fe/Mn ratios to calculate the pyroxenite-derived melts proportion in the primary magmas, the effect of high H₂O, hence f_{O_2} on the Fe/Mn ratios in olivine should also be considered.

λ Extended Data Figure 2a. What is the shallow green color in the section DL3-4-4 represent? Altered minerals or else?

λ Overall, the manuscript lacked the rigour and scrutiny of the dataset that I would expect for Al-in-olivine thermometry analysis. I have checked the data for olivine and spinel analysis, and I found the compositions of some of the olivine-spinel pairs do not satisfy the experimental calibration range: spinel Cr# (0 to 0.69), spinel Fe³⁺ content (0 to 0.11 atoms per 4 oxygens). Coogan et al. (2014) suggested that use of the thermometer substantially outside of the calibration range should be avoided. Moreover, the detailed error analysis and error propagation was lacking for both Al-in-olivine and Fe/Mg in olivine-melt thermometry analysis.

λ For the melt inclusion data of the ELIP picrites, the paper of Ren et al. 2017, GCA should be evaluated.

λ For the Al-in-olivine thermometry of the ELIP picrites, a recently paper published by Xu and Liu., 2016. Lithos should be evaluated.

λ For calculating the Fe/Mg olivine liquidus temperature of the ELIP basalts, the authors assumed the crystallization pressure of cpx phenocrysts is 1.3 GPa. However, the compositions of these cpx have been measured by the authors. Therefore, the crystallization pressure of cpx phenocrysts can be independently calculated according to the Equation 30 of Putirka. (2008), which uses the compositions of coexisting liquid and cpx.

Reviewer #2 (Remarks to the Author):

Mantle hydration and the role of water in the generation of large igneous provinces by Liu, Xia, Kuritani, Hanski, and Yu.

Petrological inferences about the thermal state of the mantle sources that yield Large Igneous Provinces and Ocean Islands are complicated by a number of factors that the authors of this paper address. For example, it is known that the amount of H₂O and recycled crust in the source can

potentially compromise mantle potential temperature TP estimates. In particular, water can lower olivine liquidus temperatures and TP, raising the potential of increasing melt production without the need of a hot mantle plume. Recycled crust may weigh the plume down, increasing opportunities for cooling (Trela et al., 2015; EPSL). But these factors need to be quantified.

What the authors have done is provide new geochemical and petrological data on picrites from the Emeishan Large Igneous Province which permit a quantitative way of evaluating the roles of TP, H₂O and recycled crust in their petrogenesis. To my knowledge this has not been done previously in a comparably comprehensive manner for any specific LIP occurrence. It is not easy but, with one exception, I found their methods to be reasonable. So their conclusions follow from the data and their interpretations. The authors conclude that the Emeishan LIP partially melted from a hot mantle plume that contained significant H₂O and recycled crust; high H₂O contents remain even though the uncertainty is significant. But they took the problem one step further by comparing the Emeishan results to other LIP and OIB occurrences. They show that H₂O has played an important role in Phanerozoic LIPS, and they conclude with an evaluation of the sources of the water. This is very a thoughtful and important paper, worthy of publication in Nature Communications. My comments are minor and should help expedite a quick revision.

Minor Comments:

1) The authors estimate the amount of recycled crust using Fe/Mn following the method of Sobolev et al. (2007). I think this is generally reasonable. However, the authors might play it safe by considering their estimates to be uppermost bounds based on the recent paper by Matzen et al. (2017; Nature Geoscience).

2) The authors discuss at the end of their paper the introduction of H₂O into the mantle by subduction in the past. Jun Korenaga has given the mantle hydration problem a lot of thought, most recently: Korenaga J, Planavsky NJ, Evans David AD. 2017. Global water cycle and the coevolution of the Earth's interior and surface environment. Phil. Trans. R. Soc. A 375: 20150393.

Claude Herzberg,
July, 2017.

Reviewer #1 (Remarks to the Author):

Traditionally, the formation of large igneous provinces (LIPs) is interpreted as the product of mantle plumes. Because of its higher mantle potential temperature than the normal mantle, upwelling of a mantle plume will cross the dry peridotite solidus at more deeper depth than that beneath the MORB mantle, thus can generate higher volumes of magmas (assuming a similar final melting depth, this will lead to a significantly wider melting interval). Petrology, isotope and trace element geochemistry of MORBs and OIBs and continental basalts show that the Earth mantle is highly heterogeneous as the consequence of recycling of crustal material by subduction or delamination into the convection mantle. For this reason, the presence of fusible fertile components (e.g., pyroxenite) and volatiles (e.g., CO₂ or H₂O) in the mantle source should also remarkably reduce the mantle solidus, therefore capable for the genesis of the LIPs.

In this manuscript, Liu et al. present, to my knowledge, the first Fourier transform infrared (FTIR) spectrometry analysis of clinopyroxene phenocrysts and other geochemical results of picrites in the Emeishan LIP. The authors show that the estimated primary magmas of these picrites have very high water contents, which is similar to the island arc basalts, but significantly higher than those for MORB and OIB. Based on these results, they proposed that mixing a low H₂O/Ce pyroxenite endmember with a hydrous peridotite endmember could explain their geochemical features. At last the authors suggested the deep mantle during the Phanerozoic is widely hydrated. The H₂O data for the ELIP picrites is new, but the explanation of these data has some flaw as listed below, and thus a major revision is required before it can be accepted for publication.

□ Lines 17-18. It is puzzling that why such a high H₂O content do not affect trace element compositions of the magmas, since both island arc basalts and most CFBs with high H₂O contents usually display negative Nb-Ta anomaly (e.g., reviews by Plank et al. 2014, EPSL for IAB and by Wang et al. 2016, Lithos for CFB).

You likely to the paper by Plank et al., (2013, EPSL 364, 168-179). It is common for IABs to display a high water content and negative Nb-Ta anomaly due to the large extent of metasomatism of the mantle wedge, from where the IABs are derived, by the dehydrated fluids with/without

some contribution of marine sediments.

Dehydration and/or hydrous partial melting in the subduction zone, especially in the stability field of rutile, would take away much more fluid-mobile elements, i.e. LILE, and some LREE than HFSE, leading to relative enrichment in HFSE in the residual slab. Many recent experimental and natural investigations have shown that within such a residual subducted oceanic slab, many minerals can carry a large amount of water into the deep mantle (see the review by Ohtani, 2015 CG). If such water is released in the deep mantle after long-term heating by the ambient mantle or a plume, the resulting fluids or hydrous partial melts would not show Nb and Ta depletion. Alternatively, wadsleyite in the mantle transition zone would contain much more water but significantly less trace elements, as suggested by Sobolev et al. (2016), the involvement of such mantle transition zone materials would make the resulted partial melts show high water contents but non-arc-like trace element patterns. From the work of Wang et al. (2016), the estimated water content for the ELIP magmas is 4-5 wt.%, and for other CFB, it is 1-2 wt.%. However, none of the most primary magma of global CFBs show typical arc-like trace elements (as shown in our Extended Data Figure 7). Actually, in Figure 5 of Wang et al. (2016), there are many samples displaying Nb/La higher than 1, especially in the Deccan, Emeishan, Columbia River and Karoo CFBs.

Recently, an independent melt inclusion study on the volatile contents of the Siberian Trap meimechite (Ivanov et al. 2017, <https://goldschmidt.info/2017/abstracts/abstractView?id=2017001546>) showed that the water content of the primary magma would be up to 3.88 wt.%, while the trace element patterns are rather like those of typical OIBs (Sobolev et al., 2009). Also, the Pechenga ferropicrites with OIB-like trace element characteristics were relatively high in water (Hanski & Smolkin 1995; *Lithos* 34,107-125). Thus, a high water content in a primitive magma can be associated both with depletion and enrichment in HFSE.

□ Lines 44-45. “A fast kilometer-scale uplift prior to eruption of ELIP basalts” is only an explanation, not an observational result. Some researchers have argued against this interpretation (e.g., Ukstins Peate. 2008, *Nature Geosciences*).

Mentioning of the large-scale crustal uplift in the ELIP in the early manuscript as part of the description of the geologic background of our research targets was to show that the ELIP has likely experienced a considerable extent of decompression. Of course, neglecting the questioning

of this conclusion is our fault. We have modified the description in the revised manuscript.

□ Lines 46-47. Wang et al. 2016, Lithos has independently estimated the water contents (4.0-5.6%) in the primary magmas of the ELIP picrites by using the liquid line of descent (LLD) for a large dataset of the ELIP basalts. Their results are similar to the data of the current manuscript. Therefore, these previously work should be evaluated, although the previous authors did not report direct analysis of H₂O content in the ELIP picrites.

In the revised manuscript, we have compared our results with this work.

Actually, the water contents of the primary magma in global LIPs WERE NOT calculated based on LLD. Instead, they were calculated based on the H₂O/Ce and Rb/Nb, Ba/Nb ratios, and the Ce content for the magma equilibrium with olivine of Fo=91. It should be especially noted that this H₂O/Ce-Rb/Nb (and others) relationships were calibrated from a dataset of global oceanic arc basalts. In Wang's work, the estimated water content for the parent magma (~4.4 wt.%) when plagioclase was among the liquidus phases (MgO of bulk rock is < 8wt.%) is similar to their estimated water content for the primary magma (4.0-5.6 wt.%) (equilibrium with olivine with Fo=91), which needs further explanation. In addition, the most primary magma in ELIP DO NOT have arc-like trace element signatures (Hanski et al., 2010), which makes the direct application of H₂O/Ce-Rb/Nb to ELIP not so straightforward.

□ Line 58. I checked the data, and I found the LOI values for these basalts are not low, ranging from 3.9% to 5.3%, thus the alteration effect should be examined.

We have checked the effect of alteration on both the major and trace elements. All the major element oxides, except Na₂O, are correlated with less-mobile TiO₂; Na₂O is positively correlated with MgO. These observations show that alteration did not change significantly major element concentrations. For trace elements, only the fluid-mobile elements, like Pb, Rb, Ba, and Sr, have poor correlation with Nb, an immobile element, which indicates little changes by alteration.

□ Lines 60-63. Based only on the high Ni contents and high Fe/Mn ratios of olivines, the authors proposed a significant contribution of melts derived from a pyroxenite source. However, this is not always the case. Because all the melting temperature, pressure, and melt or olivine composition factors will control on the Ni contents and Fe/Mn ratios, e.g., a recently 2017 paper published in Nature Geosciences by Matzen and coauthors suggested that the high Ni contents and high Fe/Mn ratios of olivines can also be explained by peridotite melts at different temperatures and pressures.

So I suggest the authors to recheck all these factors separately.

Yes, the recent study of Matzen et al. (2017) really gave us an alternative explanation for the variation of the Ni content and Fe/Mn ratio of olivine. However, this explanation could not be regarded as evidence against the presence of pyroxenite in the mantle source of basalts. As stated by the authors at the end of their paper, “*Some of the scatter, and some of the systematic behavior could be due to compositional heterogeneities of peridotites and/or the presence of lithologically distinct recycled components (for example, pyroxenites) in their sources*”. As suggested by Matzen et al. (2017), the deeper the partial melts segregate from their mantle source (corresponding to higher T and MgO content in the melts), the higher the Fe/Mn ratios of the olivine crystallized at shallow pressure are. According to this principle, the Turtugal picrites would show much higher Fe/Mn ratios than the MORBs, because the partial melting of their source occurred at ~7 Gpa and >1750°C (Trela et al. 2017, Nature Geoscience), much deeper than in the case of MORBs. This also applies to Archaean komatiites (Herzberg 1992). However, as shown in our Figure 3, the Fe/Mn ratios (linearly correlated with X_{px}) of olivines from all these rocks are comparable to that of MORBs. In Figure 3, there are also other cases that could not be explained by peridotite melting. For instance, the Tarim picrites that were erupted on the Tarim Craton with a lithospheric mantle thickness of >150 km display similar Fe/Mn ratios with MORBs; the different stages (post-caldera and syn-caldera) of the SRP basalts have large variable Fe/Mn ratios, which require large differences in the lithosphere thickness that was not true. In addition, for the Ni content of olivine, a simple partial melting of peridotite could not explain why the olivines in the Karoo ferropicrites have considerably higher Ni content than the Hawaii basalts at given Fo (Extended Data Figure 4a), although they have similar partial melting P-T conditions (Matzen et al., 2017, supplementary data; Figure 2 in this work). Overall, we suggest that the partial melting of peridotite at different P-T conditions alone could not explain our Fe/Mn and Ni data.

Actually, the identification of a pyroxenite source does not only depend on olivine Fe/Mn and data. The compositions of the primary magma, like the CaO-MgO relationship, could provide an additional constraint (Herzberg, 2011; Herzberg and Asimow, 2008; Herzberg and Asimow, 2015). The Dali picrites, Karoo ferropicrites and Siberian Gd picrites, all with relative high olivine Fe/Mn ratios and Ni contents, have relative low CaO contents and could not pass through the test by PRIMELT2 (the alert “pyroxenite source” are present). In contrast, the Tutugal picrites, Tarim

picrites, and komatiites with relatively low olivine Fe/Mn ratios and Ni contents could pass the source lithology test. The consistency between the olivine and bulk-rock data suggests that our olivine compositions can reflect their source lithological characteristics. However, considering the effect of pressure and temperature of the partial melting on Fe/Mn ratios, we think that the calculated X_{px} should represent a maximum estimate of the amount of pyroxenite in the source.

By the way, the very recent paper in EPSL (Howarth and Harris, 2017, 475, 143-151) shows that the olivine from the Karoo CFB Tuli and Mwenezi picrites have systematically higher Ni content and Fe/Mn ratios than that of the olivine from the Etendeka CFB Horingbaai low-Ti type picrites, although they were all erupted in regions with a thick lithosphere (>90 km). This should give additional confidence that olivine trace element characteristics could reflect those of the mantle source.

□ Lines 65-67. The authors have analyzed the chemical compositions of melt inclusion, I suggest to analyze the water content in melt inclusions of olivines as well, this can provide more convincing self-consistent results about H₂O contents in the primary magmas.

The compositional data of the melt inclusions were measured 7 years ago, not in the recent study.

Actually, the work of O'Leary et al. (2010) has shown that the method based on cpx phenocrysts could provide water estimates that are rather consistent with primary melt inclusions. However, the H₂O content in the inclusions hosted by olivine are not necessarily consistent with the results based on the cpx phenocrysts. First, many studies have called for caution to use the melt inclusions hosted by olivine to recover the primary water content of magma, because the water in melt inclusions can diffuse out through olivine very fast in the form of H⁺, especially in the case of volcanic rocks erupted subaerially (e.g. Gaetani et al., 2012, *Geology*). This indicates that it is not always safe to directly compare the melt inclusion data with the phenocryst-based results. Tian et al. (2017, *Geology*) show that the H diffusivity in cpx is much slower than that in olivine, which means that cpx could provide a more convincing record than olivine-hosted inclusions when diffusion does occur. Second, as shown in Ren et al. (2017), a large proportion of the inclusions (672 of 864) hosted in the Emeishan olivines are too evolved, with MgO <12wt.%.

□ Lines 67-70. Because such a high H₂O content of the primary magmas should significantly decrease the partition coefficients of Ca between olivine and melt, thus the crystallized olivines

should have low Ca contents (Gavrilenko, Herzberg et al. 2016, JP). However, I find the olivine Ca contents are not lowered remarkably. The author need to explain how to reconcile this discrepancy.

As stated on page 1821 of this JP paper, *Propagating uncertainties in the anhydrous parameterizations, the total uncertainty in magmatic water inferred from $DCaO^{O/L}$ and MgO is $\pm 1.4\% H_2O$ (1sd) for the low-MgO population and $\pm 1.8\% H_2O$ (1sd) for the high-MgO population.* This is a huge uncertainty, which means that a high-MgO magma with a water content up to 3.6 wt.% can have $DCaO^{O/L}$ similar to that for a magma with 0 wt.% H_2O , with equal probability. In their Appendix Table A3, for the Loihi Hawaii samples with similar bulk-rock MgO, the observed $DCaO^{O/L}$ of 0.0257 and 0.0225 lead to water content estimates of 0.1 and 1.6 wt.%, respectively. In addition, if you compare their Figure 6a (for dry magma) and Figure 7 (for calibrated wet magma), you will find that the magma with an inferred water content of up to ~3 wt.% will definitely plot within the range of dry magma used for calibration. Such a large uncertainty actually means that the relationship between the water content and Ca partitioning is rather complex.

On the other hand, there are indeed some arc lavas with a high water content, but they do not have considerably low CaO contents in their olivine phenocrysts. As shown in the following figure, while the Aeolian Arc lavas usually have relatively low Ca, the Vesuvius arc lavas and Italian arc leucitites show considerably higher Ca. The Ca content of the Dali picrite olivines overlap with the range of these hydrous arc lavas. The detailed reason for this is beyond the scope of our work.

Ca content vs. Mg# of olivine. Modified from Zamboni et al., Lithos, 2016. The Dali picrite data are from Ren et al. (2017) and this work.

□ Line 71-76. The contaminated continental crust by high MgO magmas should not be the average composition of the continental crust, but the regional crust composition in the ELIP as suggested by Sobolev et al. 2016, Nature for the Archaean Abitibi high-Mg Komatiites.

Yes, in the revised manuscript we applied the local crustal composition.

□ Lines 99-106. For estimating the mantle potential temperature, the author noted that water contents and the presence of pyroxenites in the peridotite source would significantly overestimate the calculated T_p . However, in the detailed Methods Section, the T_p were calculated using the method proposed by Herzberg and Asimow. 2015. This method is still based on dry peridotite hypothesis. The authors show that the estimated T_p and the uncertainty is ~ 1500 °C and ± 70 °C, respectively. Considering the H₂O contents in the mantle source, this value should be lower furthermore. Even so, I remained question the method for calculating T_p used by the authors if the peridotite source contains high water contents and pyroxenites.

In fact, we have evaluated the effects of both water and pyroxenite on the T_p calculation. We are sorry to cause this misunderstanding due to unclear description. In the revised manuscript, we have modified the description.

As described in the main text and Methods, because no any existing model could be applied directly to our case, we put forward two steps to estimate the T_p : 1) evaluation of the effects of water on the liquidus temperature of olivine; 2) calculation of the T_p with the relationship between this liquidus temperature and mantle potential temperature (Herzberg and Asimow, 2015), which is calibrated by peridotite-based experiments. Then we evaluate the error of applying this peridotite-based model to pyroxenite-involved case by comparing this model with partial melting experiments of pyroxenite (Extended Data Figure 10 and Supplementary Table 4). In the first step, we calculated the $T_{ol-liquidus}$ using the Al-in-olivine thermometer, which is not sensitive to the H₂O content and fO_2 , and also by the Fe-Mg exchange thermometer (Putirka et al., 2008), which can be largely affected by H₂O. As shown in Extended Data Figure 6, when the recovered water content (~ 3.6 wt.%) were incorporated into the Fe-Mg thermometer, the calculated $T_{ol-liquidus}$ from both thermometers is consistent with each other within error. This consistency implies that the water in

magma indeed depressed the liquidus temperature of the olivine. This is the consideration of the effect of water. In the second step, when we used the peridotite-based model (Herzberg and Asimow, 2015) to recover the mantle potential temperature of partial melting experiments on pyroxenite (n=110) (the real T_p in experiments were calculated with experimental temperature and pressure), we found that for 67 runs, the recovered T_p by the peridotite-based model are consistent with the experimental T_p within difference of 50°C, and for 88 runs, the recovered T_p are consistent with experimental T_p within difference of 70°C. This error would decrease if the melting source contains both peridotite and pyroxenite, which is the case for our samples. We thus suggested that the application of the peridotite-based model to a pyroxenite-involved source would recover T_p for a common mantle source, within a general maximum error of 70°C (the confidence level is 80%, not significantly lower than that represented by 2SD). In fact, in the recent work by Trela et al. (2017, Nature Geoscience), the same method has been applied to many LIPs, including ELIP, whose sources have been suggested to contain pyroxenite. This is our consideration of the effect of pyroxenite.

□ Lines 139-146. Although Sobolev et al. 2016, Nature suggested a mantle plume origin for the Archaean komatiites, Herzberg. 2017, JP suggested an alternative model for Archaean komatiites. He provided detailed petrological and mass balance modeling evidence for an early Earth carbon and water cycle. Then he suggested these rocks were generated in the asthenospheric convection mantle in the presence of volatiles such as H₂O or CO₂, without the requirement for thermal mantle plume or anomalous high mantle potential temperature.

I guess you mean the paper of Herzberg (2016, JP). Yes, Herzberg suggested the presence of volatiles, such as H₂O or CO₂, in the source of the Archaean komatiites, but he never denied the mantle plume model. In the abstract of his paper, he states that “*Depending on the extent of volatile degassing, hydrous and CO₂-rich komatiites could have formed either in mantle plumes or in ambient mantle*” and “*Hydrous and CO₂-rich komatiites formed from carbonated wetspots in mantle plume or ambient mantle later in the Archean*”. In fact, whether the wet mantle plume or hydrous ambient mantle should be accepted, depends on how we understand the extent of degassing of volatiles in komatiites. As shown in his discussion, to satisfy the conditions of melt production in ambient mantle, the primary komatiite melts should contain 3 wt.% of H₂O or 5-6

wt.% of CO₂, which are tremendously higher than the measured values for all the global komatiites so far. Actually, as shown by Sobolev et al. (2016), the water content of the melt inclusions decrease considerably only when the Fo of olivine are lower than 92, which means that degassing of the melt inclusions hosted by early crystallized olivine (Fo>92) is not significant.

Overall, given the present dataset, the mantle plume model for the Abitibi komatiites remains most reasonable, albeit the mantle source was likely hydrous as suggest in our paper.

□ Line 153. I checked the reference cited here, and I found it is not related volatile elements transported into the deep mantle.

In the revised manuscript, this reference has been substituted by Garnero et al., (2016, Nature Geoscience), which is from the same group as the previous reference and argue for the role of volatiles in the LLVPs.

□ Line 154. Although the mantle has cooled enough, based on Al-in olivine thermometer, Trela et al. 2017, Nature Geosciences has suggested an ancient mantle reservoir sufficiently hot for the Phanerozoic Tortugal lavas.

Yes, Trela's work indeed indicates that there is survival of a sufficiently hot ancient lower mantle reservoir. The existence of such hot, relatively young lavas implies that regions in the Earth's mantle with extreme temperatures are still present. However, as reviewed by Shorttle (2017, Nature Geoscience), the significance of this hot reservoir still needs further evaluation in the future in a self-consistent dynamical model for Earth's mantle, and the hot lava could be alternatively explained by that the Galapagos plume dredged material from the thermal boundary layer at the core-mantle boundary, where temperatures probably rise by more than 1000°C over just a few hundred kilometers. In other words, it still remains unclear whether this restricted occurrence could have a global meaning. On the other hand, the existence of this residual hot deep mantle does not argue against the cooling of the shallower ambient mantle since the Archaean, which has been proved by many lines of petrological evidence (Herzberg et al., 2010). It is the thermal gradient of the shallow ambient mantle that controls the dehydration efficiency during the oceanic slab's subduction.

□ Figure 3. For those mantle source with higher H₂O/Ce, their source should also have higher fO₂, which equals to higher Fe³⁺ and lower Fe²⁺. This could lead to crystallization of olivines with low Fe/Mn ratios. Thus, before using olivine Fe/Mn ratios to calculate the pyroxenite-derived

melts proportion in the primary magmas, the effect of high H₂O, hence fO_2 on the Fe/Mn ratios in olivine should also be considered.

We agree with you that that the Fe/Mn ratios of olivine would be effected by the H₂O and hence the fO_2 . But it would not be equally propagated to H₂O/Ce ratios, because a low water content does not necessarily mean low H₂O/Ce ratios, like in the case of the Abitibi komatiites, which only contain H₂O ~0.8 wt.% but has H₂O/Ce higher than 6000 (Sobolev et al., 2016). In the revised manuscript, we evaluate the potential effect of fO_2 on the Fe/Mn ratio carefully, and we suggest that the trend shown in our Figure 3 could not be explained by variation of fO_2 caused by variable H₂O.

In the following table, the water content, H₂O/Ce, fO_2 and Fe/Mn ratio are shown for the Gd picrite and meimechite in the Siberian Trap, and the Abitibi komatiites. It indicates that the X_{px} (Fe/Mn) ratios are not controlled by the water content, H₂O/Ce, or fO_2 .

	H ₂ O (wt.%)	H ₂ O/Ce	fO_2 (ΔQFM)	X _{px} (Fe/Mn)
Gd picrite	0.25	149	-1.5~-2 (Sobolev et al., 2009)	0.94
Siberian meimechite	3.88	202	+1.5 (Sobolev et al., 2009)	0.03
Abitibi komatiites	0.8	6524	-1.2 (Sobolev et al., 2016)	0.20

Extended Data Figure 2a. What is the shallow green color in the section DL3-4-4 represent? Altered minerals or else?

These materials are due to precipitation from alteration fluids, after the igneous process.

Overall, the manuscript lacked the rigour and scrutiny of the dataset that I would expect for Al-in-olivine thermometry analysis. I have checked the data for olivine and spinel analysis, and I found the compositions of some of the olivine-spinel pairs do not satisfy the experimental calibration range: spinel Cr# (0 to 0.69), spinel Fe³⁺ content (0 to 0.11 atoms per 4 oxygens). Coogan et al. (2014) suggested that use of the thermometer substantially outside of the calibration range should be avoided. Moreover, the detailed error analysis and error propagation was lacking for both Al-in-olivine and Fe/Mg in olivine-melt thermometry analysis.

Yes, it's our flaw not to filter the dataset very strictly. In the revised manuscript, we checked this carefully.

There are only four spinels having Cr# higher than 0.69, and neglecting these data does not change the basic trend between T and Mg# and the estimation of the olivine liquidus temperature at all. There are indeed many spinels in our dataset having an Fe³⁺ content higher than 0.11 (p.f.u). However, we suggest that these cannot result in unreliable estimates of olivine crystallization temperature, due to the following consideration. In the dataset of MORBs and LIPs in Coogan et al. (2014), all the Fe³⁺ contents are higher than 0.11 (from 0.127 to 0.238), and so are some of the spinels in his MORB dataset (up to 0.147). However, the calculated olivine crystallization temperatures for both MORBs and LIPs decrease with decreasing Fo, which is expected for a cooling process during magma evolution; and the resulting temperatures for LIPs are systematically higher than those of MORBs, which is consistent with the results of other thermometry. This means that some extent of Fe³⁺ elevation would not lead to a significant error in T estimation. The Fe³⁺ content of spinels in our dataset varies from 0.056 to 0.2 (less than 0.238 for the Coogan's dataset) and the calculated temperatures also decrease with decreasing Fo and increasing Mn but do not correlate with Fe³⁺ content and Fe³⁺/Fe_{total}.

For the error of the thermometers, we followed the suggestions by Coogan et al. (2014) and Putirka (2008), because we have a similar analytical error for FeO-MgO and Al₂O₃ in olivine, and followed exactly the same approach with them.

□ For the melt inclusion data of the ELIP picrites, the paper of Ren et al. 2017, GCA should be evaluated.

In the revised manuscript, the data of Ren et al. (2017) have been compared with our data.

□ For the Al-in-olivine thermometry of the ELIP picrites, a recently paper published by Xu and Liu., 2016. Lithos should be evaluated.

Actually, Xu and Liu (2016) used the previously published data of olivines and spinels (Kamenetsky et al., 2012, JP; Hanski et al., 2010) for the Al-in-olivine thermometry calculation, which has been partly (the Dali picrite data) involved in the Extended Figure 6 (labeled as *IT Hanski* in this figure).

□ For calculating the Fe/Mg olivine liquidus temperature of the ELIP basalts, the authors assumed the crystallization pressure of cpx phenocrysts is 1.3 GPa. However, the compositions of these cpx have been measured by the authors. Therefore, the crystallization pressure of cpx phenocrysts can be independently calculated according to the Equation 30 of Putirka. (2008), which uses the

compositions of coexisting liquid and cpx.

In the revised manuscript, we calculate the crystallization pressure of the cpx phenocrysts by our data with equation 30 of Putirka et al. (2008), with the result being 12.9 kbar, i.e. rather consistent with that we assumed previously.

Reviewer #2 (Remarks to the Author):

Mantle hydration and the role of water in the generation of large igneous provinces by Liu, Xia, Kuritani, Hanski, and Yu.

Petrological inferences about the thermal state of the mantle sources that yield Large Igneous Provinces and Ocean Islands are complicated by a number of factors that the authors of this paper address. For example, it is known that the amount of H₂O and recycled crust in the source can potentially compromise mantle potential temperature TP estimates. In particular, water can lower olivine liquidus temperatures and TP, raising the potential of increasing melt production without the need of a hot mantle plume. Recycled crust may weigh the plume down, increasing opportunities for cooling (Trela et al., 2015; EPSL). But these factors need to be quantified.

What the authors have done is provide new geochemical and petrological data on picrites from the Emeishan Large Igneous Province which permit a quantitative way of evaluating the roles of TP, H₂O and recycled crust in their petrogenesis. To my knowledge this has not been done previously in a comparably comprehensive manner for any specific LIP occurrence. It is not easy but, with one exception, I found their methods to be reasonable. So their conclusions follow from the data and their interpretations. The authors conclude that the Emeishan LIP partially melted from a hot mantle plume that contained significant H₂O and recycled crust; high H₂O contents remain even though the uncertainty is significant. But they took the problem one step further by comparing the Emeishan results to other LIP and OIB occurrences. They show that H₂O has played an important role in Phanerozoic LIPS, and they conclude with an evaluation of the sources of the water. This is very a thoughtful and important paper, worthy of publication in Nature Communications. My comments are minor and should help expedite a quick revision.

Minor Comments:

1) The authors estimate the amount of recycled crust using Fe/Mn following the method of Sobolev et al. (2007). I think this is generally reasonable. However, the authors might play it safe by considering their estimates to be uppermost bounds based on the recent paper by Matzen et al. (2017; Nature Geoscience).

We have considered the effect of partial melting conditions on the estimate of the amount of recycled crust, based on the work of Matzen et al. (2017).

2) The authors discuss at the end of their paper the introduction of H₂O into the mantle by subduction in the past. Jun Korenaga has given the mantle hydration problem a lot of thought, most recently: Korenaga J, Planavsky NJ, Evans David AD. 2017. Global water cycle and the coevolution of the Earth's interior and surface environment. Phil. Trans. R. Soc. A 375: 20150393.

We have considered these new results in the implications for H₂O cycling in the revised manuscript.

Claude Herzberg,

July, 2017.

REVIEWER COMMENTS:

Reviewer #1:

The author have addressed our comments in detail. I suggest acception.